# Seemingly Redundant Modules Enhance Robust Odor Learning in Fruit Flies

**Haiyang Li[1*]**    **Liao Yu[2*]**    **Qiang Yu[3†]**    **Yunliang Zang[1,4†]**

[1]Academy of Medical Engineering and Translational Medicine, Tianjin University, Tianjin, China
[2]School of Mathematical Sciences, Beihang University, Beijing, China
[3]School of Artificial Intelligence, Tianjin University, Tianjin, China
[4]Xiamen Intretech Inc, Xiamen, Fujian, China
`haiyangli@tju.edu.cn`
`yuliao_16@buaa.edu.cn`
`yuqiang@tju.edu.cn`
`yunliangzang@tju.edu.cn`

## Abstract

Biological circuits have evolved to incorporate multiple modules that perform similar functions. In the fly olfactory circuit, both lateral inhibition (LI) and neuronal spike frequency adaptation (SFA) are thought to enhance pattern separation for odor learning. However, it remains unclear whether these mechanisms play redundant or distinct roles in this process. In this study, we present a computational model of the fly olfactory circuit to investigate odor discrimination under varying noise conditions that simulate complex environments. Our results show that LI primarily enhances odor discrimination in low- and medium-noise scenarios, but this benefit diminishes and may reverse under higher-noise conditions. In contrast, SFA consistently improves discrimination across all noise levels. LI is preferentially engaged in low- and medium-noise environments, whereas SFA dominates in high-noise settings. When combined, these two sparsification mechanisms enable optimal discrimination performance. This work demonstrates that seemingly redundant modules in biological circuits can, in fact, be essential for achieving optimal learning in complex contexts. The code is available at: `https://github.com/L-0cean/Fly-SNN`.

## 1 Introduction

Biological redundancy is commonly observed in the brain, where different regions, pathways, or mechanisms can perform similar functions [1–4]. Different motifs of biological redundancy may exist; for instance, different modules may have evolved to fulfill similar roles, ensuring robust neural functions under pathological conditions [5–8]. Alternatively, these mechanisms may be required to achieve near-optimal learning in more complex environments. Comparative analyses of the roles of putatively redundant modules in learning can clarify how the brain adapts and, in turn, inform theories that guide neuromorphic design [9, 10]. For instance, core computational features of the fly olfactory circuit have motivated the FlyLoRA architecture [11], which enhances task decoupling and parameter efficiency.

The fly olfactory system, a canonical cerebellum-like circuit, is a tractable model for dissecting neural computation owing to its relative simplicity and well-characterized anatomy and function

---

[*]Equal contribution.
[†]Corresponding author.

39th Conference on Neural Information Processing Systems (NeurIPS 2025).

[12–18]. Two motifs—lateral inhibition (LI) and spike-frequency adaptation (SFA)—are prominent in the mushroom body and related circuits and both are known to shape neuronal responses [19–21]. LI is mediated by inhibitory interneurons that constrain the spatial spread of excitation and sharpen population representations [22, 23]. SFA reflects spike-triggered neuronal adaptation currents that accumulate during sustained stimulation, producing a progressive reduction in firing rate and emphasizing stimulus onsets and changes [24, 25].

Prior work has extensively characterized the roles of LI and SFA in neural information processing. LI drives competition and winner-take-all dynamics [26], increases population sparseness [27], enhances input contrast [28], and decorrelates activity patterns [29], it also facilitates subtractive or divisive gain modulation and strengthens regularization [30]. SFA functions as a nonlinear high-pass filter [31], promoting intensity-invariant coding [32], encoding changes in input statistics [33], sparsifying temporal responses, and forming short-term memory with non-stored retrieval (distinct from synaptic memory) [25].

Although the roles of LI and SFA in sparsifying neuronal responses are well documented, their relative contributions to learning—particularly under naturalistic conditions—remain less well characterized. Both mechanisms are expected to transform odor inputs into spatially and temporally sparse codes, attenuate noise, and enhance pattern separation [34–38](Figure 1). However, how they shape noisy odor representations and support discrimination learning across different noise regimes is not well understood. From an AI perspective, designing noise-robust classifiers is a classical challenge; elucidating how a fly learns to classify noisy odors may inform the development of robust, biologically inspired algorithms [39].

In this work, we developed a fly olfactory circuit model to investigate the roles of LI and SFA in odor discrimination tasks under varying noise conditions that simulate complex environments. Our main finding is that LI enhances learning performance in low- and medium-noise conditions, but this benefit gradually diminishes and may reverse when odors become noisier. In contrast, SFA consistently improves odor learning regardless of noise levels. LI tends to be more effective in low- and medium-noise environments, while SFA shows superior performance under high-noise conditions. When combined, the enhancement effects of these two mechanisms can be added up to achieve the optimal performance. These results suggest that seemingly redundant modules may be selectively recruited to optimize learning under different noisy conditions.

## 2   Method Overview

We developed a spiking neural network model of the fly olfactory circuit. The input layer consists of 50 olfactory receptor neurons (ORNs) that transduce odor stimuli into spikes. Each ORN projects to a corresponding projection neuron (PN), yielding 50 PNs. We also included local interneurons (LNs), which receive excitatory input from ORNs and provide lateral inhibition onto PNs. PN activity is relayed to 2,000 Kenyon cells (KCs) in the mushroom body; each KC samples inputs from approximately six PNs on average. Finally, KCs converge onto mushroom body output neurons (MBONs), which serve as readout units; in our task configuration, each MBON corresponds to an odor class being learned [40, 41].

The fly olfactory pathway exhibits three canonical features that are critical for discrimination [42]: large expansion (PN → KC), sparse connectivity, and sparse coding. Here, we focus on the role of spatiotemporally sparse spike coding, which conventional ANN models cannot capture; other architectural and connectivity parameters were constrained to experimentally observed values.

A schematic of the network and its putative role in odor discrimination is shown in Figure 1. After the KC stage, the sparsification mechanisms can transform odor responses in ways that facilitate odor discrimination. For example, they may preserve the original inter-class separability while increasing intra-class compactness (top), increase inter-class separation while keeping intra-class compactness unchanged (middle), or simultaneously achieve high intra-class compactness and improved inter-class separability (bottom). Regardless of the specific transformation, these changes are supposed to enhance the decision boundaries for odor discrimination.

Synaptic plasticity was restricted to the KC→MBON connections; all other synapses were fixed. Details of the training procedure and plasticity rule are provided in the Learning Algorithm section.

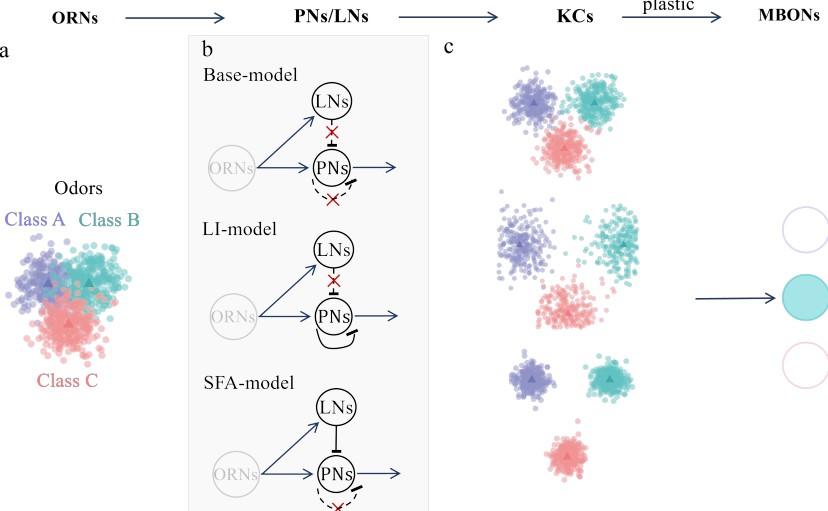

Figure 1: **Schematic of the fly olfactory circuit model.** Odor inputs are sensed and encoded by ORNs. After passing through the ORNs, odor-triggered spikes in PNs can be shaped by two factors: LI caused by LNs and SFA by inherent adaptation currents. Neuronal spikes in KCs may subsequently show different variation patterns to favor odor discrimination in MBONs.

**Input Data.** We adapted the odor-discrimination dataset from [43]. For each odor class $i$, we define a prototype ORN response vector $\mathbf{I}_i \in \mathbb{R}^{50}$. Its components are independently sampled from a uniform distribution: $I_{i,j} \sim \mathrm{U}(0,1)$ for $j = 1, \ldots, 50$. An individual sample from class $i$, denoted $\mathbf{I}_i^{(k)}$, is generated by adding an additive noise vector to the prototype: $\mathbf{I}_i^{(k)} = \mathbf{I}_i + \mathbf{Y}^{(k)}$, where every component of the noise vector $\mathbf{Y}^{(k)} \in \mathbb{R}^{50}$ is independently sampled from a zero-mean Gaussian distribution, $Y_j^{(k)} \sim \mathcal{N}(0, \sigma_{\text{noise}}^2)$. In accordance with the non-negativity of ORN firing rates, we applied element-wise clipping so that $\mathbf{I}_i^{(k)} \geq 0$.

**Neuron Model.** All neurons were modeled as leaky integrate-and-fire (LIF) units with a soft reset. The membrane potential of neuron $j$ in the layer $X$ evolves according to:

$$\tau_{\text{m}}^X \frac{\mathrm{d}V_j^X(t)}{\mathrm{d}t} = -V_j^X(t) + I_{\text{input},j}^X(t) + I_{\text{bias},j}^X + I_{\text{SFA},j}^X(t) + I_{\text{LI},j}^X(t) \tag{1}$$

Where $\tau_{\text{m}}^X$ is the membrane time constant. $I_{\text{input},j}^X(t)$ denotes the input current to neuron $j$ in layer $X$, arising from odor-evoked stimulation or synaptic drive from presynaptic neurons; $I_{\text{bias},j}^X$ is a constant bias that sets a baseline drive (used primarily in PNs and LNs to provide a small excitatory input); $I_{\text{SFA},j}^X(t)$ is a spike-triggered adaptation current that depends on the neuron's recent spiking history and typically exerts a net inhibitory effect; and $I_{\text{LI},j}^X(t)$ is the lateral inhibitory current from LNs onto PNs, scaled by LN spiking activity.

In the model, a spike is emitted when $V_j^X(t)$ reaches the threshold $V_{\text{th}}^X$. Upon spiking, the membrane potential undergoes a soft reset: $V_j^X(t^+) = V_j^X(t) - V_{\text{th}}^X$. For simplicity, we used the same membrane time constant and threshold across layers ($\tau_{\text{m}}^X = \tau_{\text{m}}$, $V_{\text{th}}^X = V_{\text{th}}$), except in the readout (MBON) layer, where the threshold was set to 1.5 times higher. Given the dense KC $\to$ MBON connectivity, this higher threshold mitigates excessive spiking driven by the convergence of inputs from thousands of KCs.

**LI Mechanism.** LI onto PNs is mediated by LNs and modeled as an inhibitory current driven by an exponentially decaying trace of LN spiking. The inhibitory current onto PN $j$ is:

$$I_{\text{LI},j}^{\text{PN}}(t) = \sum_k w_{\text{LN}\to\text{PN},jk} T_k^{\text{LN}}(t) \tag{2}$$

Where $w_{\text{LN}\rightarrow\text{PN},jk} \leq 0$ are inhibitory synaptic weights, and $T_k^{\text{LN}}(t)$ is the spike-triggered trace of LN $k$ that integrates recent activity and decays over time:

$$\tau_{\text{trace\_LN}}\frac{\mathrm{d}T_k^{\text{LN}}(t)}{\mathrm{d}t} = -T_k^{\text{LN}}(t) + S_k^{\text{LN}}(t) \tag{3}$$

$$S_k^{\text{LN}}(t) = \sum_l \delta(t - t_{kl}^{\text{LN}}) \tag{4}$$

with $\tau_{\text{trace\_LN}}$ the decay time constant, $S_k^{\text{LN}}(t)$ the spike train of LN $k$, and $t_{kl}^{\text{LN}}$ the $l$-th spike time. To compensate for the added inhibition and maintain comparable PN firing rates across conditions, we apply a modest increase in PN input drive when LI is enabled.

**SFA Mechanism.** SFA was implemented in PNs, LNs, and KCs as an inhibitory, spike-triggered current driven by an exponentially decaying state variable. For neuron $j$ in layer $X$,

$$I_{\text{SFA},j}^X(t) = w_{\text{SFA}}^X A_j^X(t) \tag{5}$$

$$\tau_{\text{SFA}}^X \frac{\mathrm{d}A_j^X(t)}{\mathrm{d}t} = -A_j^X(t) + S_j^X(t) \tag{6}$$

$$S_j^X(t) = \sum_l \delta(t - t_{j,l}^X) \tag{7}$$

where $A_j^X(t)$ integrates the recent spiking history and decays with time constant $\tau_{\text{SFA}}^X$; $w_{\text{SFA}}^X \leq 0$ sets the strength (sign) of the inhibitory adaptation current; $S_j^X(t)$ is the spike train of neuron $j$ (Dirac delta representation), and $t_{j,l}^X$ denotes the $l$-th spike time.

To maintain comparable firing rates across conditions when SFA is enabled, we applied a small positive bias current to PNs and LNs.

**Learning Algorithm.** In our model, classification is based on the average membrane potential of the MBONs over a fixed evaluation window rather than on spike counts. Membrane potentials vary more smoothly than discrete spike trains and thus provide a more stable signal for gradient-based optimization [44, 45]. The complete training procedure is outlined in Algorithm 1.

---

**Algorithm 1** Simplified SNN Training with Adaptive Mechanisms for Olfactory Tasks

---

1: **Input:** Training dataset $D_{\text{train}}$, Learnable weights $W_{\text{KC}\rightarrow\text{MBON}}$, Non-learnable parameters $P$.
2: **for** epoch = 1, 2, ..., num_epochs **do**
3:     **for** each batch $(x_{\text{batch}}, y_{\text{batch}})$ in $D_{\text{train}}$ **do**
4:         **Phase 1: Temporal Forward Simulation**
5:         Initialize all SNN states $H_0$. Set $H_{\text{hist}} = []$.
6:         **for** step $t = 0$ to num_steps - 1 **do**
7:             $I_t = \text{Input}(x_{\text{batch}}, \text{step})$                       $\triangleright$ Get input current
8:             $H_{t+1} = F(I_t, H_t, P)$        $\triangleright$ SNN forward pass and state update
9:             Record MBON membrane potentials in $H_{\text{hist}}$.
10:         **end for**
11:         **Phase 2: Backpropagation and Weight Update**
12:         $\hat{Y}_{\text{batch}} = G(H_{\text{hist}})$                     $\triangleright$ Process MBON output
13:         loss = $L(\hat{Y}_{\text{batch}}, y_{\text{batch}})$              $\triangleright$ Cross-Entropy Loss
14:         loss.backward()         $\triangleright$ Perform Backpropagation Through Time
15:         Optimizer.step()             $\triangleright$ Update $W_{\text{KC}\rightarrow\text{MBON}}$
16:     **end for**
17: **end for**

---

For input sample $i$, we recorded each MBON's membrane potential over the evaluation window $[t_s, t_e]$ and computed its time average. Let $N_{\text{MBON}}$ be the number of MBONs. The mean potential for MBON $j$ and the corresponding vector across MBONs are

$$\bar{V}_{\text{m},i}^j = \frac{1}{t_e - t_s}\int_{t_s}^{t_e} V_j(t)\,\mathrm{d}t, \quad \bar{\mathbf{V}}_{\text{m},i} \in \mathbb{R}^{N_{\text{MBON}}} \tag{8}$$

We convert $\bar{\mathbf{V}}_{\mathrm{m},i}$ to class probabilities using a $\mathrm{softmax}$ function and train the network with a cross-entropy loss:

$$p_i^j = \mathrm{softmax}(\bar{\mathbf{V}}_{\mathrm{m},i})_j \tag{9}$$

$$L_i = -\sum_j^{N_{\mathrm{MBON}}} y_i^j \log(p_i^j) \tag{10}$$

Where $y_i^j$ is the one-hot target for sample $i$.

We optimized the learnable KC $\rightarrow$ MBON synaptic weights $W_{\mathrm{KC}\rightarrow\mathrm{MBON}}$ using backpropagation through time (BPTT), unrolling the network over the evaluation window. Because LIF spikes arise from a step nonlinearity with zero derivative almost everywhere, we used a surrogate gradient during the backward pass. Specifically, we replaced the derivative of the Heaviside with a smooth arctan-based surrogate:

$$\sigma'(u) \approx \frac{k_1}{1 + (k_2 u)^2} \tag{11}$$

where $k_1, k_2 > 0$ are scaling constants.

Training uses Adam with L2 weight decay (weight regularization) to improve generalization. We also employed a ReduceLROnPlateau scheduler that monitors classification accuracy on a held-out set and reduced the learning rate by a factor $\alpha = 0.2$ if accuracy did not improve for $n = 10$ epochs.

Each model was trained for 100 epochs using mini-batches of size 256. After each batch, the loss was computed, gradients were backpropagated through time, and parameters were updated using the optimizer.

## 3 Experiments

**Dataset and models:** We generated an odor dataset comprising 30,000 training samples and 10,000 test samples, each drawn at random as described above. For most simulations, we evaluated three network configurations: (i) the Baseline model, which includes neither LI nor SFA; (ii) the LI model, which adds LI onto PNs; and (iii) the SFA model, which implements SFA in PNs, LNs, and KCs. In Figure 4, we also simulated the full model with both LI and SFA.

**Experimental settings:** Simulations were implemented in Python (snnTorch) and run on an NVIDIA A800 GPU. Network dynamics were simulated with a 1-ms time step; each trial comprised a 10-ms pre-stimulus baseline followed by a 30-ms odor presentation. All neurons shared a membrane time constant of 10 ms. Firing thresholds were 0.8 for PNs, LNs, and KCs, and 1.2 for MBONs. Connectivity featured sparse PN $\rightarrow$ KC projections (each KC received inputs from 6 PNs; fixed weight 0.3) and fully connected KC $\rightarrow$ MBON synapses initialized uniformly in $[0, 0.08]$. LI used a 5-ms LN trace time constant, and SFA used a 50-ms adaptation time constant. Models were trained for 100 epochs (batch size 256) with Adam (initial learning rate $1.0 \times 10^{-4}$). Performance was evaluated from the time-averaged MBON membrane potentials over the stimulus window.

## 4 Results

We evaluated our fly olfactory network model on a noisy odor discrimination task by incorporating LI and SFA mechanisms. The results show that these two seemingly redundant mechanisms play complementary roles in learning across varying noise conditions, thereby optimizing odor discrimination.

### 4.1 Odor discrimination in noise-free conditions

Our results show that the fly olfactory circuit model learns odor discrimination effectively (Figure 2). We first examined discrimination in noise-free conditions. In the baseline model—without LI or SFA—the discrimination accuracy reaches 91.7% for $1,000$ odor classes. Although accuracy decreases as the number of classes increases, it remains 74.72% for $10,000$ odor classes. These

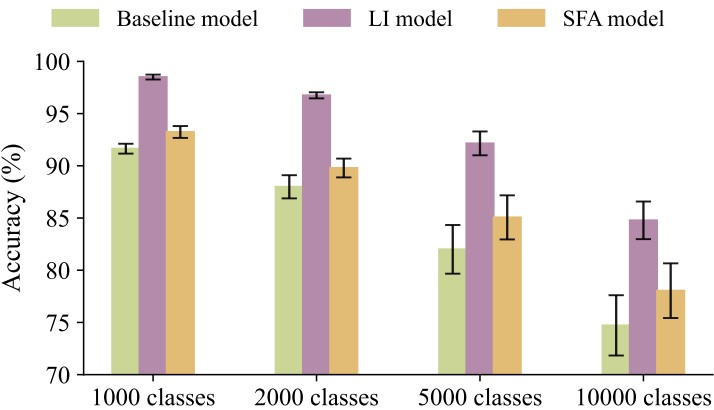

Figure 2: **Odor discrimination accuracy for fly olfactory circuit variants in a noise-free setting.** Performance is shown for three configurations——Baseline, LI, and SFA models——as a function of the number of odor classes (1,000–10,000).

findings indicate that the circuit's other features confer strong pattern-classification capacity even without explicit sparsifying mechanisms in the circuit [42, 46].

Both LI and SFA improve odor discrimination relative to the Baseline model across all tested numbers of classes. By comparison, the LI model achieves significantly higher accuracy than the SFA model under the same conditions.

## 4.2 Odor discrimination in noisy conditions

Table 1 presents a comparative analysis of discrimination performance under varying noise intensities for the Baseline, LI, and SFA models, with odor classes ranging from $1,000$ to $5,000$. At low- and medium-noise levels—defined as noise intensity (N.I.) $< 0.20$ for $1,000$- and $2,000$-class odor discrimination, and N.I. $< 0.15$ for $5,000$-class odor discrimination—the results are consistent with the noise-free context (LI model $>$ SFA model $>$ Baseline model), although overall test accuracy decreases. These findings indicate that sparsification of neuronal spikes in KCs via LI is more effective than SFA in enhancing odor discrimination under no- and low-noise conditions. However, when odor inputs become noisier (N.I. $\geq 0.20$ for $1,000$- and $2,000$-class discrimination, and N.I. $\geq 0.15$ for $5,000$-class discrimination), the SFA model consistently achieves the highest discrimination accuracy.

Table 1: **Comparative performance of the Baseline, SFA, and LI models for 1,000-, 2,000-, and 5,000-class odor discrimination under different noise ntensities.** Magenta values indicate the highest performance under each condition, violet values represent the second-best performance, and blue values denote the lowest performance.

| | | 1000 classes | | | 2000 classes | | | 5000 classes | | |
|---|---|---|---|---|---|---|---|---|---|---|
| | N.I. | Baseline | LI | SFA | Baseline | LI | SFA | Baseline | LI | SFA |
| | 0 | 91.70 | **98.30** | 93.50 | 88.85 | **96.85** | 90.90 | 82.00 | **92.15** | 85.06 |
| | 0.05 | 78.46 | **96.35** | 82.78 | 68.62 | **92.47** | 74.81 | 52.22 | **77.47** | 58.93 |
| Acc. | 0.1 | 74.61 | **91.85** | 78.26 | 61.70 | **83.07** | 67.87 | 38.65 | **57.83** | 45.18 |
| (%) | 0.15 | 74.04 | **83.63** | 79.94 | 61.14 | **71.52** | 68.54 | 34.47 | 42.15 | **43.89** |
| | 0.2 | 72.64 | 74.78 | **78.77** | 58.80 | 58.87 | **67.34** | 32.70 | 31.91 | **43.04** |
| | 0.25 | 67.32 | 62.63 | **74.34** | 52.26 | 45.84 | **61.55** | 27.74 | 22.26 | **36.94** |
| | 0.3 | 59.03 | 53.82 | **69.34** | 43.34 | 37.50 | **55.78** | 20.26 | 16.55 | **30.38** |

We systematically analyzed the impact of the strength of each sparsification mechanism—LI and SFA—on discrimination performance (Figure 3). The simulation results indicate that both facilitation effects depend on noise levels, but in distinct ways. For LI, discrimination accuracy initially improves

as N.I. increases, but then gradually diminishes and can even reverse at higher noise levels. Across most of the tested N.I. range, stronger inhibition promotes discrimination accuracy. Compared to the Baseline model, strong inhibition improves accuracy by 6.80% when N.I. = 0.0 and 18.52% when N.I. = 0.1. Notably, although LI impairs discrimination at high N.I., ($-4.5\%$ when N.I. = 0.30), the reduction is still less pronounced for stronger inhibition.

In contrast to LI, SFA consistently facilitates odor discrimination across all tested N.I. ranges. As with LI, greater SFA strength generally yields higher accuracy, with improvements of 1.80% when N.I. = 0.0, 3.65% when N.I. = 0.1, and 11.56% when N.I. = 0.30.

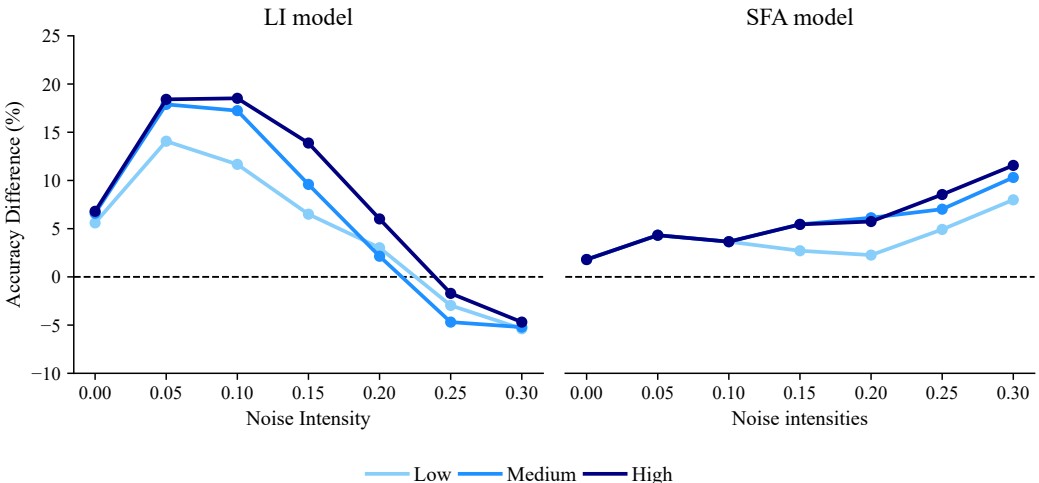

Figure 3: **Changes in discrimination performance of the SFA and LI models relative to the Baseline model across varying degrees of inhibition and adaptation under different noise intensities.** Results are shown for 1,000-class odor discrimination only. The "Low," "Medium," and "High" conditions (distinguished by color) represent increasing strengths of the respective mechanisms, achieved by systematically adjusting the relevant synaptic weights: $w_{\text{LN}\to\text{PN}}$ in Eq. 2 for LI, and $w_{\text{SFA,X}}$ in Eq. 5 for SFA. For both mechanisms, "Low," "Medium," and "High" correspond to weight increases in an approximate 1:2:3 ratio.

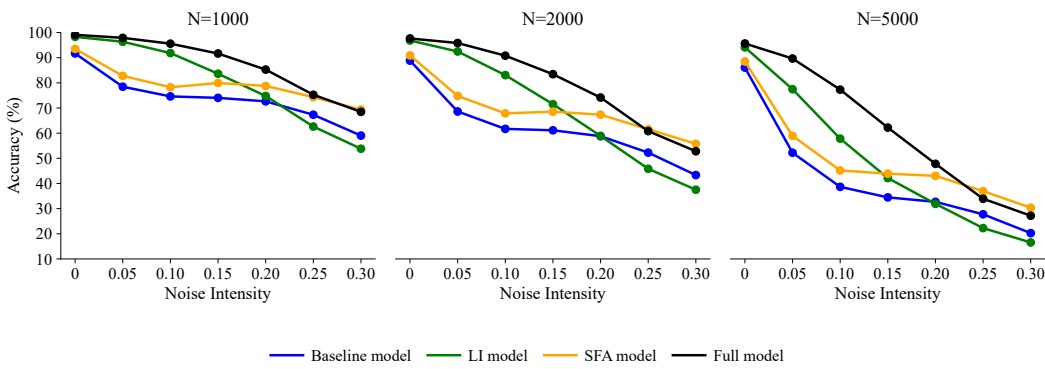

Figure 4: **Discrimination performance of the SFA model, LI model, Full (SFA + LI) model, and Baseline model under different noise intensities.** The models were tested on odor discrimination tasks with 1,000, 2,000, and 5,000 odor classes.

In addition, we investigated whether LI and SFA could be dynamically combined to achieve optimal odor discrimination learning (Figure 4). Our results show that orchestrating these two mechanisms produces higher discrimination accuracy than using either mechanism alone under low- and medium-noise levels. Only under high-noise conditions does the SFA model outperform all other models. The benefits of these two mechanisms are additive. These findings suggest that, although each mechanism

is most effective within specific noise regimes, LI and SFA can work together in a complementary manner, combining their individual effects to optimize the learning process.

## 4.3 Learning speed

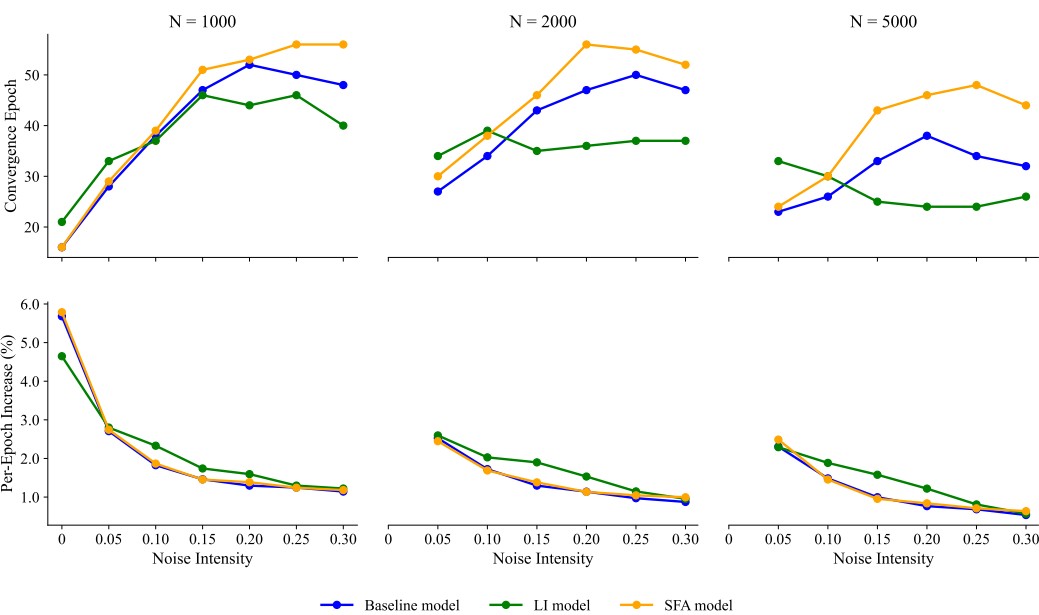

Figure 5: **Impacts of LI and SFA on model convergence speed.** The top (bottom) panel shows the number of training epochs required for convergence (the accuracy gain per epoch) for the Baseline, LI, and SFA models, plotted against noise intensity for different odor category sizes: 1,000, 2,000, and 5,000 classes. To avoid potential misinterpretations from relying solely on maximum accuracy and to objectively assess learning progress, convergence is defined as the point at which model accuracy growth plateaus. Specifically, a model is considered converged when the average improvement in accuracy over $n = 10$ consecutive epochs falls below a predefined threshold (threshold = 0.003).

To further assess the impacts of LI and SFA on learning efficiency, we provide a quantitative analysis of how these two sparsification mechanisms shape the time course of odor discrimination learning. As shown in Figure 5 (top), similar to the final discrimination accuracy values (Table 1), the number of training epochs required for convergence depends on noise level. In the low-noise range, the LI model requires more epochs to reach the defined converged state, regardless of odor category size. SFA follows LI, while the Baseline model reaches convergence fastest. The convergence speed can be explained by combining the final discrimination accuracy (Table 1) with the accuracy gain per epoch (Figure 5, bottom). In the low-noise range, both LI and SFA require higher final accuracy thresholds to converge, but their per-epoch gains show no advantage over the Baseline model, resulting in slower convergence.

In higher-noise ranges, however, the LI model exhibits the fastest convergence, followed by the Baseline model, with the SFA model consistently the slowest. In this regime, the LI model generally maintains an advantage over the Baseline model in per-epoch accuracy gain, explaining its faster convergence (Figure 5, top). When N.I. = 0.3, the per-epoch accuracy gain is similar across models; however, SFA requires a higher final accuracy threshold to reach convergence compared to both the Baseline and LI models (Table 1). Consequently, LI reaches the converged state first, the Baseline model second, and the SFA model last.

These results suggest that, depending on the noise level, learning speed—alongside discrimination accuracy—may be an important factor in determining which sparsification mechanism is recruited during noisy odor discrimination learning.

### 4.4 Sensitivity analysis

#### 4.4.1 Sensitivity to noise types

The previous results were obtained by simulating the odor discrimination task with Gaussian noise. To assess the generalizability of our findings, we evaluated the effects of the sparsification mechanisms under different noise environments. In addition to Gaussian noise, we simulated noise generated by an Ornstein–Uhlenbeck (OU) process, which captures the temporal correlations often observed in natural odors. Odor samples with OU noise were generated using a procedure analogous to that used for Gaussian noise.

The results, summarized in Table 2, show that the effects of LI and SFA on discrimination performance with OU noise are generally consistent with those observed using Gaussian noise. LI achieves the highest discrimination accuracy at low- and medium-noise levels, whereas SFA is most effective at higher noise intensities. The main difference is that, within the tested noise range, LI—like SFA—consistently improves the discrimination of noisy odors compared to the Baseline model, rather than impairing performance at high noise levels, although its benefit gradually diminishes. Overall, these findings confirm that the complementary effects of LI and SFA are robust across the tested noise types.

Table 2: **Comparative performance of the Baseline, LI, and SFA models for 1,000-class odor discrimination under different OU noise intensities.**

|  | N.I. | Baseline | LI | SFA |
|---|---|---|---|---|
| | 0.1 | 86.02 | **97.98** | 88.66 |
| | 0.3 | 78.66 | **96.43** | 82.55 |
| Acc. (%) | 0.5 | 76.01 | **93.10** | 78.85 |
| | 0.9 | 74.54 | **87.84** | 78.50 |
| | 1.5 | 67.55 | 69.72 | **77.49** |

#### 4.4.2 Parameter sensitivity analysis

To further verify the reliability of our simulation results, we conducted a comprehensive parameter sensitivity analysis. Three key hyperparameters—random seed, learning rate, and batch size—were varied while keeping all other experimental conditions fixed to assess their impacts on odor discrimination.

Our tests reveal that:

- Varying the random seed typically changed accuracy by less than 1.3%.

- Adjusting the learning rate to 0.5–2.0 of its default value resulted in accuracy changes generally under 1.6%.

- Scaling the batch size to 0.5–2.0 of its standard value produced accuracy differences typically below 3.0%.

These results demonstrate that, within reasonable variation ranges, model performance is highly robust to hyperparameter choices. Importantly, across all parameter variations, our core conclusions remain unchanged: the LI model achieves optimal performance in low- and medium-noise environments, whereas the SFA model performs best under high-noise conditions. This analysis confirms that our main findings are reliable and independent of specific hyperparameter settings, providing a strong foundation for the interpretations presented in this study.

## 5 Conclusion

In this paper, we present a computational demonstration of how two distinct neural sparsification mechanisms—circuit-level LI and neuronal-level SFA—provide complementary advantages for noisy odor discrimination. Both mechanisms, well-documented in biological neural systems, actively shape and refine spatiotemporal neuronal responses. Our simulation results reveal a noise-dependent

recruitment pattern: LI is preferentially engaged under low-noise conditions, whereas SFA dominates in high-noise environments. The fly can orchestrate these two sparsification mechanisms to achieve optimal discrimination performance. These findings illustrate how seemingly redundant circuit modules in biological systems may, in fact, represent an optimized strategy for maintaining robust learning performance across diverse environmental conditions. This study sits at the interface of computational neuroscience and spiking neural networks, leveraging AI methods to investigate learning processes in the fly olfactory circuit. While this interdisciplinary approach is enriching, it also carries several limitations, as discussed below.

## 6   Limitations

We present a computational model of the fly olfactory circuit and investigate the roles of LI and SFA in odor discrimination learning. While our findings offer valuable insights, several limitations point to promising directions for future research.

First, regarding the training and test datasets, our simulations used artificially generated odors [43]. Although the odor generation method was experimentally inspired and provided a controlled environment, future work should validate odor discrimination performance using more naturalistic and physiologically realistic odor stimuli.

Second, our current model assumes that plasticity is confined to synaptic connections between KCs and MBONs. The synaptic update rule follows backpropagation through time, which is generally considered biologically implausible. While we believe this does not alter our conclusions, future studies should explore biologically plausible learning algorithms. Moreover, as in most cerebellum-like circuit models, the synaptic weights between the input and hidden layers (ORN–KC synapses) are fixed [18, 47]. These connections may also undergo other forms of plasticity, such as unsupervised learning via Oja's rule [48]. Exploring these possibilities would provide a more comprehensive understanding of how learning unfolds across the entire circuit.

Finally, although our parameter robustness analysis and exploration of different noise types support the generalizability of our findings, the parameter ranges and noise characteristics examined remain limited. Broader investigations into diverse environmental conditions and increased model complexity could further strengthen the conclusions.

## 7   Acknowledgement

This work was supported by the National Key Research and Development Program of China (2023YFF1204200), the National Natural Science Foundation of China (62476197, 12372060, 92370103, 62176179), and the Xiaomi Foundation. The funders had no role in study design, data collection and analysis, decision to publish, or preparation of the manuscript.

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

# A Discrimination Accuracy over Training Epochs under Different Noise Intensities

The main text primarily presents the final discrimination accuracy values under specific experimental conditions. To better illustrate the learning dynamics, this section includes results showing the time course of recognition accuracy for the three models (Baseline, LI, and SFA models) as they learn to discriminate 1,000 odor classes under varying noise intensities (N.I. = 0.1, 0.2, and 0.3), as shown in Figure 6.

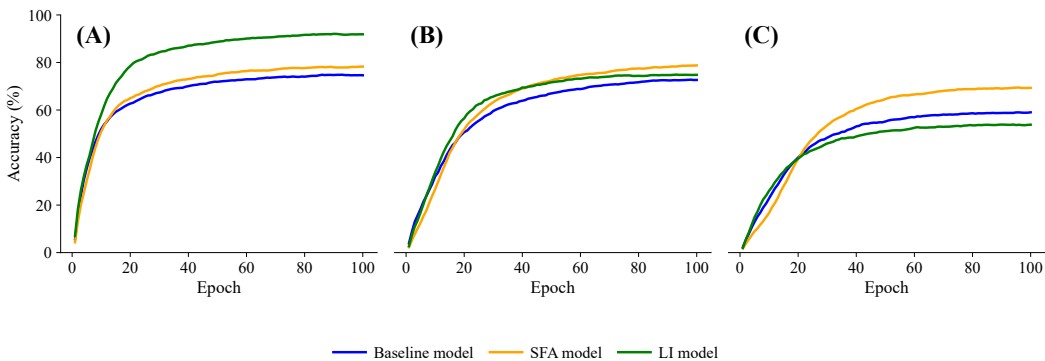

Figure 6: **Evolution of discrimination accuracy over training epochs under different noise intensities.** The subfigures show results for: (A) N.I. = 0.1, (B) N.I. = 0.2, and (C) N.I. = 0.3.

# B Model Performance with Enhanced Strength of Input Signal

The experiments described in the main text imposed certain limits on the strength of the input odor signals to better isolate performance differences attributable to the LI and SFA mechanisms. Under conditions with a large number of odor categories and substantial noise interference, this constraint led to relatively low absolute accuracy values across all models. In this section, we demonstrate that a moderate increase in input signal strength can significantly enhance overall discrimination performance.

Table 3: **Comparative performance of the Baseline, SFA, and LI models for 10,000-class odor discrimination under different noise intensities with enhanced strength of input signal.**

|         | N.I. | Baseline | LI | SFA |
|---------|------|----------|-------|-------|
|         | 0    | 99.82    | 99.99 | 99.97 |
|         | 0.05 | 98.46    | 99.90 | 99.45 |
|         | 0.1  | 96.87    | 98.59 | 98.14 |
| Acc. (%) | 0.15 | 90.13   | 86.58 | 93.27 |
|         | 0.2  | 69.80    | 60.79 | 81.31 |
|         | 0.25 | 45.21    | 34.17 | 60.07 |
|         | 0.3  | 25.94    | 18.40 | 37.45 |

Table 3 summarizes the final discrimination accuracy of the Baseline, LI, and SFA models after increasing the input signal strength. Enhancing the input signal strength led to substantial accuracy improvements for all models across the tested conditions. For example, with 10,000 odor classes and noise = 0.1, the accuracy of the Baseline model rose from 21.80% to 96.87%.

