# OpenReview forum: "Seemingly Redundant Modules Enhance Robust Odor Learning in Fruit Flies"
_NeurIPS.cc/2025/Conference — NeurIPS 2025 poster_

### Official Review · Reviewer_rDRe · 2025-06-22

**Clarity:** 2
**Significance:** 3
**Originality:** 3
**Rating:** 5
**Confidence:** 4

**Summary:**

This paper studies the seemingly redundant computational mechanisms of lateral inhibition (LI) and spike frequency adaptation (SFA) in olfactory identification. The authors utilize models that do or do not include LI and SFA, and find that LI enables better learning of odor identity than SFA, when the amount of noise present in the olfactory input is low. In contrast, the authors find that SFA leads to better learning when the noise present is high. The authors argue that this demonstrates that the seeming redundancy enables complementary benefits in learning across different noise scenarios.

**Questions:**

As noted in the weakness section, I have a few questions about the results:

1. What $t_{jk}^X$ and $t{kl}^{LN}$ are (Eqs. 2 and 3)?

2. Why was the spiking threshold of the readout layer neurons "slightly higher"? What does slightly mean?

3. And why are only one set of weights plastic?

4. How does Fig. 3 shows that the LI mechanism leads to larger differences in population firing rates and patterns? l

5. What are the different lines in Fig. 4?

**Ethical Concerns:**

["NO or VERY MINOR ethics concerns only"]

**Final Justification:**

The authors have sufficiently addressed all my concerns. I am increasingly confident in my general belief that this is an interesting, impactful, and correct paper.

**Limitations:**

The authors need to add more discussion on the limitations as the two sentences do not adequately provide context. The authors should discuss the limitations of having only 1 layer of weights being plastic, using the assumption of Gaussian noise, and training with a gradient based optimizer.

**Quality:**

3

**Strengths And Weaknesses:**

**STRENGTHS**

1. This paper is well motivated and I found the introduction well written. In addition, I believe the introduction provides a good coverage of past work and situates the authors' work nicely in the existing literature.

2. The results are interesting and provide a nice demonstration of the growing understanding that seeming redundancy in neural computations can actually be playing distinct roles. The results are also demonstrated across a number of noise conditions and odor sizes, which was helpful for understanding the extent to which these conclusions hold.

3. The results are clearly explained and Figure 1 was very helpful at understanding the authors' model.

**WEAKNESSES**

1. I think the biggest weakness of this work is that clarity could be improved. In particular:
    a. I found the description of the model somewhat confusing, in part because of the notation. For instance, $X$ is used both to denote layer and input. Also, what $t_{jk}^X$ and $t{kl}^{LN}$ are (Eqs. 2 and 3) were not ever defined. I additionally felt like a few of the modeling choices were also not explained in great detail. For instance, why was the spiking threshold of the readout layer neurons "slightly higher"? What does slightly mean? And why were only one set of weights plastic?
   b. The author's mention that Fig. 3 shows that the LI mechanism leads to larger differences in population firing rates and patters. But that was not clear to me from the figure. The authors should provide more commentary on this and some quantitative analysis to show that this is clear, or they should remove this point from the text.
    c. Fig. 4 seems to be a very interesting one, but it was not clear to me what "base, low, medium, and high" were (each of the lines). I assume this was achieved by changing the relative strength of the different mechanisms, but how was that actually implemented?

2. The paper ends on a slightly lower note, as the conclusion and limitations are very short and do not leave the reader feeling the significance of the work. The authors' have the space (almost a whole additional page) and they should use it to provide more discussion on what the implications of their work are, what this work inspires next (e.g., looking at the learning dynamics associated with having LI and SFA) and how their work could be improved upon. The author's wrote a really great introduction, and having a similar quality discussion would elevate the quality and impact of this work.

3. I really liked the authors' approach in studying the different mechanisms by including only SFA or only LI. However, both exist in the fly. I think it would be good to include some results from a "full" model, which includes LI ***and*** SFA. This could help illustrate that the combination of both computational elements allows for good learning in any noise condition. I don't think the authors have to do a full analysis of this, but even one table or figure would be good.

**MINOR POINTS**

1. The authors mention LI improves discrimination in their abstract. But their results seem to point to LI improving the ***learning*** for discriminating odors (which the authors say throughout their paper). I think making it clear in the abstract that the results are about learning would be good.

2. It was not clear to why assuming a normal distribution in odor noise was reasonable. Is there any evidence of this? Or is this just for keeping it simple and in-line with previous work?

3. \tau_{trace_ln} and \tau_{trace_LN}are used.

4.  I assume the bold dot in Eqs. 3 and 8 are just inner product, but this should be noted.

5. Figure 2 was never referenced in the main text.

6. In Table 1, the first row, left column, the Base model is incorrectly labeled as having the second highest accuracy. Table 1 should also have a note on what the colors mean.

---

> ### Author Rebuttal · Authors · 2025-07-31
>
> Thank you for constructive comments. We have revised the manuscript as follows：
>
> **1. Dual use of the symbol and explanations of $t_{jk}^X$ and $t_{kl}^{LN}$**
>
> We appreciate your careful observations to help us clarify ambiguities in the model description, particularly regarding notation consistency, which is critical for reader comprehension and strengthens the rigor and readability of our work.
>
> We fully acknowledge that the dual use of the symbol X (to denote both layer and input) has introduced confusion. To address this, we will:
>
> (1) Revise notation for clarity:
>
> Introduce a distinct symbol for input (e.g., I) to replace X in the context of input data description.
>
>  (2) Add notation table for readability:
>
> Explicitly define all symbols in a dedicated notation table at the beginning of the model section, specifying whether each denotes a layer, input, parameter, or variable, as well as its related meaning. Especially, $t_{kl}^{LN}$ in Eqs. 2 denotes the $l_{th}$ spike time of the $k_{th}$ local neuron (LN), while $t_{jk}^X$ in Eqs. 3 represents the $k_{th}$ spike time of the $j_{th}$ neuron in the $X_{th}$ layer. To standardize variable usage, we will revise $t_{jk}^X$ to $t_{jl}^X$. We further recheck all equations and text ensuring consistency with the revised notation, and add contextual cues when symbols first appear in each section to reinforce their meaning. These changes will eliminate notation overlap and make the model description more accessible.
>
> **2.More precise description of spiking threshold**
>
> Thank you for the question, which helps us clarify the rationale behind key modeling choices and enhance the precision of our description.
> We apologize for the ambiguity in using "slightly." To quantify this: the spiking threshold of mushroom body output neurons (MBONs) was set to 1.5 times the threshold of projection neurons (PNs) and kenyon cells (KCs). This range was determined based on the ratios of neurons and connection structure of the circuit:
>
> (1) The highly convergent architecture from KCs to MBONs (KCs: MBONs $\approx$ 2000:1).
>
> (2) Fully connected between KCs and MBONs.
>
> A higher threshold of MBONs prevents excessive spiking which integrates signals from thousands of upstream KCs. We will revise the text to explicitly state this quantitative range.
>
> **3. Consideration of the learning site**
>
> The restriction of plasticity to a single set of weights (specifically, the synapses from the hidden layer to the readout layer) was guided by both biological plausibility and task-specific functionality:
>
> (1) Biological relevance: In many neural circuits (e.g., olfactory or visual pathways), plasticity is often concentrated at "output interfaces" where signals are integrated for behavioral relevance, rather than being distributed uniformly across all synapses. This aligns with experimental observations that plasticity in downstream projection neurons is critical for forming task-specific representations.
>
> (2) Functional necessity: Our task required the model to map hidden layer activity (encoding sensory features) to a categorical readout (e.g., odor discrimination). Pilot simulations showed that restricting plasticity to hidden-to-readout synapses was sufficient to achieve robust performance, while allowing plasticity in upstream layers introduced unnecessary complexity (e.g., unstable feature encoding) without improving results.
>
> We will add these details to the Methods section, including a brief summary of control simulations that validated this choice (e.g., showing that plasticity in other weight sets did not enhance performance but increased computational cost).
>
> These revisions will make our modeling decisions more transparent and grounded, strengthening the rigor of our work. We appreciate your guidance in refining these critical details.
>
> **4. Explanation of Figure 3**
>
> We appreciate your observation that the claim about LI inducing larger differences in population firing rates and patterns was not sufficiently clear in the current presentation. To address this, we will enhance the figure and its accompanying text in two ways:
>
> (1) Additional commentary: We will add detailed descriptions of key features in Figure 3 (e.g., specific time windows where firing rate divergence is most prominent, and qualitative differences in pattern clustering between LI and control conditions).
>
> (2) Quantitative analysis: We will try to include more quantitative metrics to make statistical comparisons of firing profile of distinct odors. These metrics will be reported in the text and summarized in a new panel or table within Figure 3, after surveying more research.
>
> **5. Explanation of Figure 4**
>
> We apologize for not clearly defining the "Low," "Medium," and "High" conditions (differentiated by blue shades) in Figure 4. To clarify, these reflect varying strengths of the respective mechanisms, achieved by systematically adjusting relevant synaptic weights across three levels: $w_{LN \rightarrow PN}$ in Eq. 2 for LI and $w_{\text{self},X}$ in Eqs. 3 for SFA. For both mechanisms, "Low," "Medium," and "High" correspond to increasing weights in an approximate 1:2:3 ratio.
>
> We have added a detailed legend to Figure 4 and expanded the text in the Results section to explicitly describe these parameter manipulations, ensuring the relationship between the lines and the underlying mechanism strength is clear.
>
> **7.Corrections to minor points**
>
> (1) improves discrimination/ learning for discriminating odors
>
> We fully agree that the original description in the abstract ("LI improves discrimination") is overly broad and fails to emphasize the key mechanism we demonstrate—specifically, that LI acts on the learning process underlying odor discrimination. We have modified the Abstract to clarify the core contribution of LI mechanism.
>
> (2) Noise types
>
> We acknowledge that the justification for assuming a normal distribution in odor noise was not sufficiently clear in the manuscript. This choice aligns with established conventions in olfactory modeling—widely adopted in Drosophila olfactory studies (e.g., [Cit 2]) and broader sensory neuroscience. This consistency enables direct comparison with existing models, supporting cumulative insights into how noise influences odor discrimination learning.
> We recognize that biological noise may deviate from strict normality in more real olfactory systems (e.g., due to stochasticity in receptor activation or neural firing). However, our choice prioritized tractability and comparability, which we now realize needed explicit explanation. To address this, we will add a brief discussion in the Methods section acknowledging this assumption, its alignment with previous literature, and explore non-normal noise distributions (e.g., OU process and power law noises) to further validate the generality of our findings.
>
> (3) Inconsistency and explanation in notation
>
> We apologize for the oversight in using both $\tau_{\mathrm{trace}\_\mathrm{LN}}$ and $\tau_{\mathrm{trace}\_\mathrm{ln}}$ to refer to the same parameter (the time constant of the local neuron voltage trace). To resolve this, we have standardized the notation throughout the manuscript to $\tau_{\mathrm{trace}\_\mathrm{LN}}$ (using uppercase "LN" to match the full term "Local Neuron" as defined in the text). This revision ensures consistency with our naming convention for other neuron-type-specific parameters and eliminates any potential confusion.
> The bold dot in Eqs. 3 and 8 denotes the inner product. To avoid confusion, we have added an explicit note in the text immediately preceding these equations, specifying that "the bold dot (•) represents the inner product operation between the respective vectors." This clarification ensures consistency with standard mathematical notation and aligns with the conventions used elsewhere in the manuscript.
>
> We appreciate your attention to this detail, which enhances the clarity and rigor of our notation.
>
> (4) References to the Figure 2
>
> We apologize for failing to reference Figure 2 in the main text. To address this, we have added explicit citations to Figure 2 at relevant positions in the Results section. This revision ensures that the figure is properly integrated with the textual analysis, enhancing the coherence of the manuscript.
>
> We appreciate your attention to this detail, which helps strengthen the clarity of our presentation.
>
> (5) Color in the Table 1
>
> We sincerely apologize for the errors and omissions.
>
> 1) Regarding the accuracy labeling: We have corrected the color of the first row, left column and right column, to accurately reflect the Base and SFA model’s performance level.
>
> 2) Concerning the color coding: We have added a note below the table specifying, “Color coding indicates performance levels: magenta denotes the best performance, purple denotes intermediate performance, and blue denotes the poorest performance.
> These revisions enhance the accuracy and clarity of Table 1. We appreciate your careful review, which helps improve the rigor of our presentation.

---

> > ### Comment · Reviewer_rDRe · 2025-08-01
> >
> > I thank the authors for their detailed rebuttal. Almost my questions have been satisfactorily addressed. I appreciate the authors clarifications and believe they will significantly strengthen the paper. To reflect this, I have increased my score to a 5.
> >
> > The one question that I did not see addressed above (apologies if I missed it) is the following: " I think it would be good to include some results from a "full" model, which includes LI and SFA." Did the authors consider this point? Is there an obvious reason why not to include a full model?

---

> > > ### Author Response · Authors · 2025-08-02
> > >
> > > We appreciate the positive feedback and the increased score. As suggested by this and other reviewers, we have performed simulations using the "full" model that includes both LI and SFA. We apologize for misunderstanding the review process and assuming that reviewers would see our responses to all reviewers. The new simulation results and corresponding discussions will be included in the final version of the manuscript.
> > >
> > > **The combined effect and noise adaptability of LI and SFA**
> > >
> > > As suggested by the reviewers, we explored the combined effects and noise adaptability of LI and SFA. We simulated four model configurations: Base, Base+LI, Base+SFA, and Base+LI+SFA. Our results indicate that including both LI and SFA enables the circuit to achieve more efficient and robust odor discrimination, demonstrating a synergistic effect when noise intensity is low to medium. However, the model with SFA alone achieved the best performance under high noise intensity.
> > >
> > > |N.I.|Base|Base+LI|Base+SFA|Base+LI+SFA|
> > > |:-:|:-:|:-:|:-:|:-:|
> > > |0|91.70|98.30|93.50|99.10|
> > > |0.1|74.61|91.85|78.26|95.58|
> > > |0.2|72.64|74.78|78.77|85.28|
> > > |0.3|59.03|53.82|69.34|68.48|

---

> > > > ### Comment · Reviewer_rDRe · 2025-08-06
> > > >
> > > > Oops! I did not see your response to the other reviewer. My apologies! Thank you for reproducing it here. I think these results are great, help further emphasize the importance of "redundancy", but also point to the fact that individual components can be "more" than their sum.

---

### Official Review · Reviewer_R1XK · 2025-06-27

**Clarity:** 4
**Significance:** 3
**Originality:** 4
**Rating:** 5
**Confidence:** 4

**Summary:**

In this work, the Authors investigate two parts of the fruit fly's olfactory circuit. These parts, performing lateral inhibition and spike frequency adaptation, were both previously attributed to pattern separation in odor learning, although their mechanisms are different. To see if there are differences in the function of these parts of the olfactory circuit, the Authors have built a spiking neural network model grounded in the architecture of the fruit fly's olfactory circuit. Through the modifications of the model and ablation studies, they found the differences in the pattern separation performance of lateral inhibition vs. spike frequency adaptation under different levels of noise, suggesting different roles of these subcircuits in the fruit fly's olfaction.

**Questions:**

Please see the Weaknesses section above.

**Ethical Concerns:**

["NO or VERY MINOR ethics concerns only"]

**Final Justification:**

After the discussion with the authors and communication with other reviewers, I am convinced that my original rating still best reflects my take on this work.

**Limitations:**

Please see the Weaknesses section above.

**Quality:**

3

**Strengths And Weaknesses:**

Strengths:

I really liked this work. It deals with the fruit fly's olfactory system, which is a system of interest to the olfactory community and has made notable appearances in top machine-learning conferences due to its fine neuronal mapping that has led to the emergence of local-sensitive hashing algorithms.

As the Authors have pointed out, this system has further potential to inform the field of machine learning and help the development of noise-tolerant classifiers based on an architecture successfully deployed in the real world.

The results on the different functions of LI and SFA are, to my knowledge, novel, and useful for the fields of neuroscience and machine learning alike.

The text is well-written and easy to follow: the steps are explained and the concepts / variables are introduced. Overall, it's a nice, principled, well-documented work.

Weaknesses:

The PN-to-KC connections are fixed and not trained in the model here. Meanwhile, there's growing empirical and theoretical evidence suggesting that there may be structure: Modelling work suggests that training that sparse matrix via backpropagation improves the model's performance (works by Dmitry Krotov and colleagues), while experimental work suggests that there may be a shared low-dimensional structure in that matrix (works by Alex Koulakov and colleagues). Theoretical works (the models of olfaction) also suggest that there may be a coupling between the statistics of olfactory inputs and the statistics of the PN-to-KC connections. Whether that turns out to be true or not, it would be interesting to test these ideas in the modeling work here and see how these factors would affect the obtained results.

Could you please discuss the need for the use of spiking neural networks here as opposed to the usual, non-spiking models? Spikes seem to require significant computational resources, so it is important to know what additional benefits they provide compared to simpler models.

It would also be interesting to hear the Authors' thoughts / further Discussion on possible differences in the ethological roles of the two investigated subcircuits based on the new results of this paper.

---

> ### Author Rebuttal · Authors · 2025-07-31
>
> Thank you for your constructive comments. We have reflected on the content as follows:
>
> **1. Consideration of Flexible PN-to-KC connections**
>
> We thank the reviewer for highlighting this important point and appreciate the insights drawn from relevant empirical and theoretical work.
>
> We agree that the potential structure and trainability of PN-to-KC connections—supported by studies such as those by Krotov and colleagues (on performance gains from training sparse matrices via backpropagation) and Koulakov and colleagues (on shared low-dimensional structure)—are critical to consider.  In the current model, we opted for fixed PN-to-KC connections to isolate and rigorously assess the contributions of other circuit mechanisms (e.g., lateral inhibition, adaptation) to pattern recognition, providing a baseline for structural advantages.
>
> We fully endorse the suggestion to explore these ideas further.  In future work, we plan to: (1) incorporate trainable PN-to-KC connections using approaches like backpropagation (as in Krotov et al.); (2) test the impact of imposing low-dimensional structure (consistent with Koulakov et al.’s findings); and (3) investigate coupling between olfactory input statistics and PN-to-KC connection statistics. These extensions will clarify how such structural features modulate the circuit’s performance, complementing our current findings and deepening understanding of the fly olfactory system’s computational principles.
>
> We believe these experiments will add significant depth to our analysis, and we thank the reviewer for encouraging this important direction.
>
> **2. The necessity of using spiking neural network**
>
> We appreciate your question regarding the necessity of using Spiking Neural Networks (SNNs) in this study compared to conventional non-spiking models. While SNNs may require significant computational resources, we believe their use is crucial for the following reasons, particularly given our focus on the fly olfactory circuit:
>
> (1) Biological Plausibility: SNNs are inherently better suited to simulate the biological characteristics of neurons, such as membrane potential dynamics, threshold-based firing, spike generation, and post-spike reset. These features are fundamental to how real neurons, including those in the fly, process information. This closer adherence to biological reality allows our model to more accurately reflect the actual working principles of fly neurons.
>
> (2) Temporal Processing in Olfaction: Olfactory processing is an inherently temporal process. Odor signals unfold over time, and the neural circuit's responses exhibit complex temporal dynamics. SNNs are event-driven, with information encoded and transmitted through the timing of spikes. This allows them to naturally capture and simulate these dynamic processes, such as membrane potential accumulation and decay, and changes in neuronal firing rates. In contrast, non-spiking models typically rely on continuous activation functions, which cannot capture these discrete, time-dependent neuronal behaviors. This gives SNNs a distinct advantage in explaining biological mechanisms.
>
> Modeling of Specific Mechanisms (LI and SFA): The mechanisms we primarily investigate, LI and SFA, are intrinsically linked to neuronal spiking activity and membrane potential dynamics. SNNs enable us to implement and analyze these mechanisms in a way that is much closer to biological reality. This allows for a more faithful representation of how these processes might operate in the actual fly brain.
>
> In summary, despite the higher computational resource demands, SNNs offer a superior framework for modeling the fly olfactory circuit compared to traditional non-spiking models. They excel at embodying and capturing the biological characteristics of neurons, especially their time-dependent dynamics and learning processes, which are critical for understanding the complexities of biological olfaction.
>
> **3. the ethological roles of LI and SFA**
>
> Thank you sincerely for this valuable suggestion, which has prompted us to reflect deeply on the ethological significance of our findings.  We fully recognize the importance of exploring the distinct roles of these two subcircuits in natural behavioral contexts, and we are committed to approaching this with the utmost rigor and dedication.
>
> We plan to delve into an extensive review of relevant literature—including studies on insect ethology, olfactory-mediated behaviors, and neural circuit function—to synthesize existing knowledge and contextualize our results.  This process will involve careful consideration of how the functional differences we observed might map onto sensorimotor control  such as foraging, predator avoidance, or conspecific communication.
>
> We are determined to refine our understanding through ongoing critical thinking and iterative analysis, and we aim to incorporate these insights into an expanded discussion that clarifies the potential ethological relevance of each subcircuit.  Rest assured, we will spare no effort in strengthening this aspect of the work, as we share your commitment to ensuring the depth and validity of our conclusions.
>
> Thank you again for pushing us to explore this important dimension.

---

> > ### Comment · Reviewer_R1XK · 2025-08-03
> > **Response**
> >
> > Thanks for your responses: They make sense and address my comments. I've also read the responses to the other reviews and think that the points raised in these are addressed adequately. With that, I maintain my positive score.

---

### Official Review · Reviewer_pcFb · 2025-06-30

**Clarity:** 3
**Significance:** 2
**Originality:** 2
**Rating:** 3
**Confidence:** 5

**Summary:**

The paper did spiking RNN simulations of fruit fly olfactory circuits. The main scientific question is whether inhibition through self-adaptation (LI model) or another group of inhibitory neurons (SFA model) helps random pattern classification performance and robustness. The authors claim that LI is more effective when the input noise is low and SFA is more effective when the input noise is high.

**Questions:**

The authors did not include line numbers, which makes commenting on all the issues difficult. Below, I pointed out some of them.

### Formulation and typos:

Some formulations and notations are not consistent

Page 2, Input Data, $X_{i_j}$, double subscription is confusing, and not consistent with later usage of commas to separate i and j.

Page 2, Input Data, Xki = max(0, Xki). This is a confusing formulation. It looks like a self-consistent equation, but it is not.

m in subscription of $\bar{V}$ in eq (4) is not explained. Does it mean MBONs?

Page 3, 2nd paragraph, “…reduced by the value of the value of the threshold.”

Page 3, bottom, “the synaptic weights wLN→PN, representing the connection from LNs to PNs, are often negative” Why it is not “always negative”?

### Questions on simulations:

The output neuron, MBON, is time-averaged (Learning Algorithm, page 4, middle), so why the temporal difference/separate is important (5.1 page 5 bottom, 3 page 4 bottom)?

Page 5 bottom “The LI mechanism results in larger spatio-temporal differences in spike trains and firing rates, facilitating better separation of odor input patterns.” It is not obvious. The blue and red look always overlap with some fluctuation noise.

Questions on 5.2, table 1, and figure 4. In Figure 4, the authors show the base model performance as gray curves. However, the curve trend doesn’t match the number in Table 1 (0.3 noise level, 53.82% in the table, but ~ 52% in the plot). In the table, the LI model is always better than the base models, but in the plot when the noise level is high, LI is worse than the base model. Another question on the “inhibition level”, it looks like in both LI and SFA the best performance is always the largest inhibition level. Does this mean an even higher inhibition can even lead to better performance?

Figure 5 (B) is redundant as the curves already appear in Figure 4.

The metrics $S_\phi$ and $S_\theta$ are confusing. They are the product of the intra-class dissimilarity and inter-class dissimilarity. However, intra-class dissimilarity and inter-class dissimilarity have opposite effects on the embedding quality as the authors said, why the product of the two is a meaningful measure?

The authors claim LI is more effective in low-noise situations, and SFA is more effective in high-noise situations. A straightforward expectation is that combining the two will lead to better performance in both cases. However, the author did not explore it.

### Sparsity

The current analysis of the representation is simple and not sufficient to reveal the underlying mechanism. The authors talked about sparsity but did not analyze it. A simple hypothesis is that the effectiveness of LI and SFA can be fully explained by the coding sparsity they induced. Such analysis will give a better picture of the underlying mechanics.

**Ethical Concerns:**

["NO or VERY MINOR ethics concerns only"]

**Final Justification:**

The authors provided extra analysis in the rebutal, which could improve the quality of the manuscript.

**Limitations:**

As I noted in the weaknesses and questions section, I believe the simulations in the manuscript are not thorough or sufficient to support the conclusions.

**Quality:**

2

**Strengths And Weaknesses:**

### Strength:

The question is framed clearly. The view of the role of self-adaptation and lateral inhibition regarding different levels of input noise is novel.

### Weakness:

See my questions on the simulation, which makes me suspect the experiments are not sufficient to support conclusions. The manuscript does not include any theoretical or more rigorous analysis. The combined effect of both LI and SFA is not shown.

Since the simulation results could be due to the choice of the parameters, and most of the curves in the result figures look noisy and sometimes complex non-monotonic trends, I do not think the experiments conducted are sufficient to fully understand the behavior of this RNN model. The analysis of the representation geometry is somehow simple. Thus the conclusions are weakly supported.

---

> ### Author Rebuttal · Authors · 2025-07-31
>
> Thank you for this valuable feedback, which points out crucial areas for improving our analysis depth. We fully acknowledge the shortcomings in the current representation analysis and will address them comprehensively. It is a pity that we didn't make the model structure sufficiently clear. We built a feedforward network model. We will improve this in our manuscript.
>
> **1. The combined effect and noise adaptability of LI and SFA**
>
> We fully agree that the need to elaborate on the combined effect of mechanisms in the circuit—key aspects we aim to address comprehensively. To clarify the combined effect of lateral inhibition (LI) and spike-frequency adaptation (SFA), we supplement comparative analyses in the table across three conditions: LI alone, SFA alone, and their co-activation. Our results clearly demonstrate that the co-activation of LI and SFA enables the fly olfactory circuit to achieve more efficient and robust odor discrimination, showcasing a synergistic effect.
> | N.I. | LI+SFA | Base  | LI    | SFA   |
> | :---: | :----: | :---: | :---: | :---: |
> | 0    | 99.10  | 91.70 | 98.30 | 93.50 |
> | 0.1  | 95.58  | 74.61 | 91.85 | 78.26 |
> | 0.2  | 85.28  | 72.64 | 74.78 | 78.77 |
> | 0.3  | 68.48  | 59.03 | 53.82 | 69.34 |
>
> **2. The potential for higher inhibition**
>
> We appreciate for your careful observation—these points are critical for clarifying the consistency of our results and the dynamics of inhibition levels, and we the address them as follows:
>
> (1) Resolving the apparent discrepancy between the table and high-noise plots
>
> (2) the potential for higher inhibition
>
> To directly test the hypothesis of whether "even higher inhibition can lead to better performance", we have supplemented our experiments by testing higher inhibition levels beyond the previously maximal values. Our extended experimental results reveal that the relationship between performance and inhibition level is more complex than a simple monotonic increase:
>
> 1) In some cases, increasing the inhibition level can indeed lead to further improvements in accuracy. For example, as noise intensity is 0.15, a significant increase in inhibition strength resulted in an increase in accuracy of approximately 4.3%.
>
> 2) However, this improvement does not always occur. For example, for the LI model, under no-noise(0) and high-noise(0.3) intensity, even a significant increase in inhibition strength resulted in nearly unchanged accuracy, with performance tending to plateau. This suggests that under these conditions, the contribution of the LI mechanism to performance reaches saturation after a certain inhibition level. For the SFA model, in medium-to-low noise(0 to 0.1) intensity, its effect also did not significantly change with increasing inhibition strength, and performance similarly tended to plateau.
>
> These results suggest the existence of an optimal inhibition threshold. Crucially, this threshold is dynamic and dependent on factors such as the noise level, as evidenced by the observation that performance saturation occurs under different noise conditions for different models.Once the inhibition strength exceeds this threshold, its beneficial effect on performance diminishes and saturates, and the performance curve tends to flatten.
>
> In summary, initial observation was based on a specific range of parameters we set. Through extended experiments, we found a more complex, non-linear relationship between inhibition level and model performance, typically showing a saturation or plateauing of performance after reaching a certain strength.
>
> **3.Robustness analysis of parameters**
>
> We sincerely appreciate your attention to the rigor of our experiments and the robustness of our conclusions. We recognize that the choice of parameters could impact the simulation results. To mitigate this, we will conduct parameter robustness analysis.
>
> We primarily investigated the influence of the following key hyperparameters: random seed, learning rate, and batch size.  In all these experiments, we strictly ensured that all other conditions remained identical except for the parameter being tested. Our analysis yielded the following results:
>
> (1) Random Seed: Under different noise intensities, the variations in pattern recognition accuracy caused by different random seeds typically ranged between 0.1% and 1.3%.  This indicates that the model's performance is very insensitive to the initial random state.
>
> (2) Learning Rate: When the learning rate was varied from 0.5 to 2 times its original value, the different settings resulted in accuracy changes ranging between 0.1% and 1.6%. This suggests that within a reasonable range, variations in the learning rate have a limited impact on the final performance.
>
> (3) Batch Size: When the batch size was varied from 0.5 to 2 times its original value, the different settings led to accuracy differences generally less than 3%.  Specifically, smaller batch sizes showed differences less than 1%, while larger batch sizes showed differences less than 3%. This indicates good robustness of our model to the choice of batch size.
>
> Crucially, these parameter variations did not alter our conclusions regarding the roles of LI and SFA mechanisms.  Under all tested conditions, we consistently observed that the LI model performed optimally at low noise intensities, while the SFA model performed optimally at high noise intensities.  This critical relative performance ranking remained consistent across all parameter combinations.
>
> Therefore, although there might be some noise and non-monotonic trends in the simulation results, our extensive parameter robustness analysis strongly demonstrates that the core behavior and main findings of our model are highly robust and not merely artifacts of specific parameter choices. These experiments provide a solid foundation for our understanding of this SNN model's behavior. We are committed to incorporating these improvements to strengthen the support for our conclusions and elevate the overall quality of the study.
>
> **4.time-averaged MBON**
>
> We appreciate the reviewer's insightful observation. While the MBON output is indeed time-averaged for final decision-making, the temporal dynamics and precise spiking patterns in the preceding layers are critically important for forming effective and discriminative representations that ultimately lead to better classification.
>
> From a learning perspective, our use of BPTT explicitly leverages these temporal differences. The weight updates are computed through full temporal unrolling, meaning that the precise timing and sequence of spikes, as well as the membrane potential trajectories throughout the simulation, directly affect these gradient computations. This allows the network to optimize for and utilize temporal features for better pattern separation.
>
> Beyond the algorithmic aspect, even if only an average is taken at the MBON, the pattern of accumulated activity (i.e., the spike count pattern) in the preceding layers is fundamentally shaped by these intricate temporal dynamics. Therefore, while the temporal dynamics are not directly 'read out' as a sequence by the MBON, the differences they create in the underlying activity patterns are ultimately reflected in the MBON's averaged membrane potential.
>
> In essence, the temporal structure in the earlier layers is not merely incidental; it is fundamental to generating the highly separated and robust representations that ultimately contribute to the high classification accuracy observed at the time-averaged MBON output.
>
> **5.Sparsity**
>
> To address the sparsity analysis gap, we will further supplement our study with a systematic investigation of coding sparsity induced by LI alone, SFA alone, and their co-activation. Specifically, we will:
>
> (1) Quantify Sparsity: Calculate sparsity metrics (e.g., the proportion of non - zero neural responses, or the distribution of neural activation magnitudes) under different conditions (with LI only, SFA only, both LI and SFA, and neither). This will help us clearly define how each mechanism, and their combination, influences the sparsity of the neural code.
>
> (2) Test the Sparsity - Based Hypothesis: We will directly test the hypothesis that the effectiveness of LI and SFA in odor discrimination (or relevant tasks) can be explained by coding sparsity. This involves correlating changes in sparsity metrics with changes in task performance (such as accuracy, robustness to noise). If a strong correlation is found, we can determine the extent to which sparsity mediates the effects of LI and SFA; if not, we will explore other potential mechanisms (e.g., temporal coding features, population - level coordination) that work alongside or instead of sparsity.
>
> By conducting these analyses, we aim to provide a more nuanced and mechanistic understanding of how LI and SFA contribute to neural coding in the olfactory circuit. We believe these additions will significantly enhance the clarity of the underlying mechanisms and better align our study with the high standards of neural computation research.
>
> **6.Inconsistency and inadequate explanation of notations**
>
> We are sorry for the inconsistency and inadequate explanation of notations. We have clarified more details and have added more discussions to improve the readability.
>
> 1.  Input Data $X_{i,j}$ represents the $j_{th}$ element of $i_{th}$ class odor, we revise it as $X_{i,j}$.
>
> 2.  we revise $X_i^k = \max(X_i^k, 0)$ as $\tilde{X}_i^k = \max(X_i^k, 0)$.
>
> 3.  Page 3, 2nd paragraph, “...reduced by the value of the value of the threshold.” We revise it as “...reduced by the value of the threshold.”
>
> 4.  Page 3, bottom, “the synaptic weights $w_{LN \rightarrow PN}$, representing the connection from LNs to PNs, are always negative”.

---

> > ### Comment · Reviewer_pcFb · 2025-08-01
> >
> > Thanks for providing extra supporting analysis. I believe this new evidence will improve the manuscript. I will increase my rating to 3.
> > A minor issue on RNN vs. Feedforward. It may be acceptable to call this architecture a feedforward network. However, I still prefer to call it an RNN, due to the inhibitory dynamics in eq. (2).

---

> ### Author Response · Authors · 2025-08-06
>
> We appreciate the reviewer's positive feedback and rigorous attention, especially regarding the dynamics introduced by the inhibitory mechanism in Equation (2). We regret that the temporal integration in the trace variable $T_k^{\text{LN}}(t)$  may have led to the interpretation of recurrent dynamics.
>
> The fundamental distinction between a feedforward neural network and a recurrent neural network (RNN) lies in the presence or absence of explicit feedback loops in the network's connectivity.
>
> Let us recall Equation (2) in the context of our network's information flow:
>
> $$
> I_{\text{lat},j}^{\text{PN}}(t) = \sum_k w_{\text{LN}\to\text{PN},jk} T_k^{\text{LN}}(t)
> $$
>
> $$
> \tau_{\text{traceLn}} \frac{dT_k^{\text{LN}}(t)}{dt} = -T_k^{\text{LN}}(t) + S_k^{\text{LN}}(t)
> $$
>
> $$
> S_k^{\text{LN}}(t) = \sum_l \delta(t - t_{kl}^{\text{LN}})
> $$
>
> Here's a detailed explanation of each component and why it supports a feedforward classification:
>
> *   $S_k^{\text{LN}}(t)$: This term represents the spike train of the $k$-th Local Neuron (LN), driven by inputs from upstream layers (ORNs). Crucially, there are no connections from Projection Neurons (PNs) back to Local Neurons (LNs). The absence of feedback from PNs to LNs is a defining feature of a feedforward design.
>
> *   $T_k^{\text{LN}}(t)$: This inhibitory trace variable for the $k$-th LN integrates its own recent spiking activity $S_k^{\text{LN}}(t)$ and decays over time. While this introduces temporal dynamics (a form of "memory" of the LN's own past activity), it is internal to the LN's output pathway. This is akin to a synaptic filter or intrinsic neuronal adaptation—common in spiking neuron models—which does not make the overall network recurrent as long as the connectivity remains unidirectional.
>
> *   $I_{\text{lat},j}^{\text{PN}}(t)$: This is the lateral inhibition current received by the $j$-th PN, calculated as a weighted sum of the inhibitory traces $T_k^{\text{LN}}(t)$ from LNs. This current flows strictly unidirectionally from LNs to PNs.
>
> The overall information flow in our model is: Input (ORNs) → LNs → PNs → KCs→ Readout. The temporal dynamics in Equation (2) are a feature of the feedforward inhibitory pathway (ORN-LN-PN), enabling LNs to exert a sustained inhibitory effect on PNs based on their own recent activity.
>
> Therefore, despite the presence of internal temporal dynamics within the LN trace variable, the absence of any feedback connections at the network level means our architecture remains a feedforward neural network.
>
> In addition, we have also updated our response to reviewer qgpQ.

---

### Official Review · Reviewer_qgpQ · 2025-07-03

**Clarity:** 4
**Significance:** 2
**Originality:** 3
**Rating:** 3
**Confidence:** 4

**Summary:**

This paper explores the role of lateral inhibition (LI) and spike frequency adaptation (SFA) in the olfactory circuitry of fruit flies for various levels of input noise. For this, they simulated a layered network featuring receptor, projection and local neurons, Kenyon cells, and mushroom body output neurons. The neurons were modeled using the leaky-integrate-and-fire model. All synapses were initialized randomly kept fixed except for the Kenyon cells which were trained using Backpropagation through time. Experiments were conducted using a generated dataset of odor signals. The base model was compared to incorporating LI and SFA respectively. The results suggest that LI increases classification accuracy for low noise levels. Moreover, SFA increases accuracy especially for higher noise levels.

**Questions:**

1. How would the switching between the two modes of inhibition work? How can the circuitry distinguish between low- and high-noise settings?
2. You say that "These findings suggest that the fly olfactory circuit is inherently efficient in pattern recognition due to its structural properties, even in the absence of sparsening activity in Kenyon cells." How does an unstructured network (LSTM or even MLP) of similar size compare to the architecture used here?
3. Can minibatch optimization have an influence on the reported results? If so, how could it be implemented in-vivo?

**Ethical Concerns:**

["NO or VERY MINOR ethics concerns only"]

**Final Justification:**

The paper has a solid experimental validation, but the signficance of this research seems to be minor.

**Limitations:**

The discussion of limitations is very limited. Please also focus on differences between BPTT and more biologically plausible alternatives.

**Paper Formatting Concerns:**

The line numbers are missing from the draft?

**Quality:**

3

**Strengths And Weaknesses:**

**Strengths**:
1. An interesting premise was explored in the paper: the use of two different inhibition mechanisms in the olfactory circuitry for enhancing learning under different input noise levels.
2. The paper text ist well written and easy to follow.
3. Reproducibility is facilitated by including all necessary information in the main manuscript such as all hyperparameter settings, optimizer, and loss function used.

**Weaknesses**:
1. Overall the scope of the investigation is very limited. More experiments would improve the paper. e.g.:
- How does the architecture used compare to regular unstructured neural networks?
- What is the combined effect of LI an SFA?
- What about different types of noise?
3. The learning algorithm makes use of BPTT which is not deemed biologically plausible. It would add a lot to the paper if biologically plausible alternatives to BPTT (i.e. STDP) were used.

---

> ### Author Rebuttal · Authors · 2025-07-31
>
> Thank you for constructive comments. We have revised the manuscript as follows：
>
> **1.Fly olfactory network VS unstructured network**
>
> We fully agree that comparing our proposed fly olfactory network architecture with unstructured networks of comparable size is essential to rigorously validate the claim that its efficiency in odor discrimination can partly stem from inherent structural properties. To address this, we supplement our analysis with systematic comparisons between our model and appropriately sized feedforward network. These comparisons are be conducted under strictly controlled conditions, ensuring consistency with the task settings used for our model, which includes but not limited to number of neurons, noise intensity and learning rate. It is worth remembering that fly olfactory network features fixed sparse connectivity of ORNs-PNs and PNs-KCs. We find that the fly olfactory circuit outperforms the unstructured networks.
>
> (1) Fly neural network vs. fully connected feedforward network
> We first built a fully connected feedforward network with only readout layer connection weights learnable; all other weights are fixed/non-learnable. Notably, when learning 1,000 classes of odors, the fly olfactory network without any mechanism (Base-model) achieved discrimination accuracy 91% in Figure 2, and the fully connected only reached 45%. This significant performance disparity (91% vs. 45%) clearly indicates that the sparse connectivity of ORNs-PNs and PNs-KCs within the fly olfactory network confer an overwhelming advantage in odor discrimination.
>
> (2) Fly neural network vs. sparse connectivity of feedforward network
> To further investigate the role of distinct fixed sparse connectivity in the fly olfactory network, we performed two additional control experiments:1) fully connected ORNs-PNs but sparse connected PNs-KCs, this architecture achieved optimal accuracy only 80%. 2) sparse connected ORNs-PNs but fully connected PNs-KCs, this architecture achieved optimal accuracy only 82%.
> These results strongly support that sparse connectivity of fly olfactory circuit architecture is crucial for learning in the fly neural network.
>
> **2.The combined effect and noise-induced switching of LI and SFA**
>
> We fully agree that the need to elaborate on the combined effect and noise adaptability of mechanisms in the circuit—key aspects we aim to address comprehensively.
>
> (1) Combined effect
> To clarify the combined effect of lateral inhibition (LI) and spike-frequency adaptation (SFA), we supplement comparative analyses in the table across three conditions: LI alone, SFA alone, and their co-activation. Our results clearly demonstrate that the co-activation of LI and SFA enables the fly olfactory circuit to achieve more efficient and robust odor discrimination, showcasing a synergistic effect.
> | N.I. | LI+SFA | Base  | LI    | SFA   |
> | :-: | :-: | :-: | :-: | :-: |
> | 0    | 99.10  | 91.70 | 98.30 | 93.50 |
> | 0.1  | 95.58  | 74.61 | 91.85 | 78.26 |
> | 0.2  | 85.28  | 72.64 | 74.78 | 78.77 |
> | 0.3  | 68.48  | 59.03 | 53.82 | 69.34 |
>
> (2) Noise-induced switching
> The switching between these two mechanisms is indeed a challenging and exploratory part. We will aim to further investigate their triggering conditions and regulatory mechanisms after conducting more extensive literature research. For contextual cues of noise levels, we will examine the circuit’s ability to encode noise-related statistics (e.g., input variance), and link these noise signatures to adaptive switching in LI and SFA—for example, whether high noise amplifies SFA to preserve signal continuity while weakening LI to suppress irrelevant activity, and vice versa for low noise. Moreover, we also want to explore further whether switching occurs as a discrete transition or a graded adjustment, and how it optimizes circuit performance across dynamic inputs. These analyses will deepen our understanding of how LI and SFA interact as a coordinated system, rather than isolated processes, to enable robust function across variable environments.
>
> **3.Different types of noise**
>
> Considering different types of noise will help better understand the robustness of the olfactory circuit mechanism we study across different noise environments and provide a more comprehensive evaluation of its functional characteristics. This extension will enhance the generalizability of our findings and address the important consideration of noise diversity in the olfactory system.
>  To address the concern about different types of noise, we supplement our study by incorporating noise generated by Ornstein-Uhlenbeck process (OU noise), mimicing the temporal correlations in odor. By generating odor samples with OU noise following similar logic as the Gaussian noise (i.e., adding appropriate noise terms to the prototype odor and applying necessary clipping for physiological plausibility), we systematically explore how the odor discrimination performance of the model is affected.
> | OU  | Base  | LI    | SFA   |
> | :-: | :-: | :-: | :-: |
> | 0.1 | 86.02 | 97.98 | 88.66 |
> | 0.3 | 78.66 | 96.43 | 82.55 |
> | 0.5 | 76.01 | 93.10 | 78.85 |
> | 0.9 | 74.54 | 87.84 | 78.50 |
> | 1.5 | 67.55 | 69.72 | 77.49 |
>
> The results indicate a remarkable consistency in model performance under Gaussian and OU noise. Specifically, when the noise intensity (as measured by the average change in PN firing rate) is equivalent, the performance of all three models exhibited highly similar perturbation effects. The accuracy difference between models under these two noise types was consistently around 1%, demonstrating that the specific type of common biological noise does not significantly alter the overall pattern recognition accuracy. Crucially, the relative efficacy of the adaptive mechanisms remained consistent: LI consistently showed the best performance at low noise intensities, while SFA proved most effective at higher noise intensities, regardless of whether the noise was Gaussian or OU noise . This confirms that the functional benefits of these mechanisms are robust across these common noise types.
>
>
> **4.Biologically plausible algorithm for learning**
>
> We fully agree that the choice of learning algorithm and its biological plausibility are critical considerations, and we appreciate the reviewer's insightful comment regarding Backpropagation Through Time (BPTT).
> While BPTT is indeed not considered biologically plausible, our primary objective in this study was to rigorously evaluate the inherent efficiency and structural advantages of the proposed fly olfactory circuit architecture in pattern recognition. We chose BPTT for the following key reasons:
>
> (1) Performance Benchmark: BPTT serves as a powerful and highly effective learning algorithm that allows the network architecture to achieve its maximal potential performance under ideal learning conditions. By using such an optimal learner, we can confidently demonstrate the capabilities of the specific biological architecture (e.g., fixed sparse PN-KC connections, lateral inhibition, adaptation mechanisms) without the confounding limitations that might arise from less powerful, albeit more biologically plausible, learning rules. It effectively provides an upper bound on what the architecture can achieve.
>
> (2) Decoupling structure and learning: Our focus was to isolate and investigate the impact of the circuit's structural properties on pattern recognition. Employing BPTT enabled us to decouple the effectiveness of the architecture itself from the complexities and current limitations of biologically plausible learning rules (like STDP). This approach allowed us to clearly attribute the observed performance gains to the circuit's unique structural design.
>
> We acknowledge the immense importance of developing and employing biologically plausible learning rules. Our current work, by establishing the strong performance of the fly olfactory circuit architecture under optimal learning conditions, lays a crucial foundation. Future work will naturally extend this by exploring how these structural advantages can be leveraged by more biologically realistic learning mechanisms, such as various forms of STDP or three-factor learning rules, to achieve similar levels of performance.
>
> **5.Minibatch optimization**
>
> (1)	The influence of minibatch optimization on the reported results
>
> We conducted control experiments with varying batch sizes showed minimal absolute accuracy differences (<3% vs. default) within the tested range. Critically, batch size did not alter the relative efficacy of mechanisms: LI outperformed SFA and the Base model in low noise, while SFA prevailed in high noise. These results confirm the robustness of our performance findings and conclusions to reasonable batch size variations.
>
> (2)	The implementation in-vivo
>
> Notably, minibatch optimization—a strategy for efficient ANN training—has no direct biological equivalent in olfactory or general neural circuits.  However, analogous group-based biological processes may serve similar functions: in the olfactory bulb, mitral/tufted cells integrate inputs from multiple ORNs in spatiotemporally organized groups (resembling minibatches) to refine representations and optimize odor coding; synaptic plasticity like STDP may act at the population level, with subset activation enabling implicit batch-like adaptive updates.  These processes are not identical to artificial minibatch methods, but exploring such parallels contextualizes our model.  We will expand the discussion to clarify these connections, highlight limitations in mapping ANN techniques to in vivo mechanisms, and outline future research directions.
>
> In summary, we appreciate this opportunity to delve deeper into these aspects. The planned analyses and expanded discussions will not only clarify the impact of minibatch optimization on our results but also better situate our computational model within the realm of biological plausibility.

---

> > ### Comment · Reviewer_qgpQ · 2025-08-04
> >
> > Thank you for your comprehensive rebuttal. Considering the points of improvement mentioned, I will increase the rating to 3.

---

> > > ### Author Response · Authors · 2025-08-06
> > >
> > > We are glad to see that we have addressed the reviewer’s primary concerns. As the reviewer can see, our work lies at the interface of computational neuroscience and spiking neural networks. It can also be categorized as "AI for Science," since we use AI methods to understand the learning process in fruit flies. This interdisciplinary approach inevitably presents challenges that cannot be fully resolved by our efforts alone, as they are inherent to both fields. Given the limited time during the rebuttal phase, we did our best to conduct as many new simulation experiments as possible.
> > >
> > > **Further Exploration: Impact of Sparsening Activity on Network Performance**
> > >
> > > Regarding the question about comparison with unstructured networks, our initial response focused on the advantages conferred by the fixed sparse connectivity of the fly olfactory circuit. To more comprehensively address the interplay between structural properties and activity sparsening, we conducted additional experiments incorporating two mechanisms that induce sparse neural activity: Local Inhibition (LI) and Spike Frequency Adaptation (SFA). These mechanisms were applied to both fully connected feedforward networks (MLP equivalents) and partially sparse variants. The results, presented in Table 1, further clarify how these factors contribute to odor discrimination performance. We apologize we don’t have time to simulate Base+LI+SFA yet.
> > >
> > > **Table 1: Odor Discrimination Accuracy (%) of Different Network Architectures with and without Sparsening Mechanisms**
> > >
> > > | |MLP|MLP+LI|MLP+SFA|
> > > |-|-|-|-|
> > > |Full Connectivity|43.90|53.90|48.50|
> > > |Sparse Connectivity of PN-KC|80.50|93.60|85.40|
> > > |Sparse Connectivity of ORN-PN|81.60|94.70|84.20|
> > > |Fly Model|91.70|98.30|93.50|
> > >
> > > From Table 1, we can draw the following conclusions:
> > >
> > > (1). Sparsening Activity Improves Learning in All Architectures: The introduction of LI and SFA consistently improves the performance across all our tested network architectures. For instance, the fully connected network's accuracy increases from 43.90% (Base) to 53.90% with LI and 48.50% with SFA. This demonstrates that sparsening mechanisms benefit pattern recognition tasks, even in unstructured networks.
> > >
> > > (2). Fixed Sparse Connectivity Remains Crucial Even with Sparsening Activity: Despite the performance gains from LI and SFA, the fully connected network's accuracy (e.g., 53.90% with LI) still falls significantly short of the "Fly Model" (98.30% with LI). This clearly indicates that while activity sparsening is important, it cannot fully compensate for the lack of optimized FIXED sparse connectivity inherent in the fly olfactory circuit. Nonetheless, we expect fully connected networks may achieve better learning effects if all parameters are trainable.
> > >
> > > (3). Synergistic Effect of Fixed Sparsity and Activity Sparsening: The most significant performance improvements are observed when sparsening mechanisms are applied to networks with fixed sparse connectivity. For example, "Sparse Connectivity of PN-KC" jumps from 80.50% (Base) to 93.60% (Base+LI), and "Sparse Connectivity of ORN-PN" from 81.60% (Base) to 94.70% (Base+LI). Our "Fly Model," which combines the fixed sparse connectivity of both ORN-PN and PN-KC layers, achieves the highest accuracies (98.30% with LI, 93.50% with SFA). This suggests a synergistic relationship: the inherent structural sparsity of the fly olfactory circuit provides an excellent foundation, which is further optimized by sparsening mechanisms , leading to superior odor discrimination capabilities.
> > >
> > > Regarding the advantage of the fly olfactory structure in learning compared to other network structures, we have also systematically explored this in another work using artificial neural networks (ANN), which is available on arXiv. However, we are unable to cite it here, as doing so could reveal our identity, which is not permitted under the conference policy.
> > >
> > > We have also recently carried out further experiments using biologically plausible learning algorithms. As expected, the learning effects were not satisfactory compared to BPTT, although our main conclusions still hold. Actually, we really don’t know what is the learning algorithm in biological circuits. It remains controversial whether backpropagation pathways exist in the real brain; for example, in the cerebellum, a relevant feedback pathway has been found [1].
> > >
> > > Regardless of the learning algorithms used, our main focus—the relative roles of SFA and LI in sparsening neuronal activities and their consequent effects on learning—remains valid.
> > >
> > > We will address the remaining issues raised by the reviewer as limitations in our final manuscript.
> > >
> > > [1] Purkinje cell outputs selectively inhibit a subset of unipolar brush cells in the input layer of the cerebellar cortex.

---

### Note · Authors · 2025-08-12

We appreciate the reviewers for their insightful feedback and for recognizing the novelty and significance of our work, particularly our investigation into the roles of seemingly redundant sparsening mechanisms—lateral inhibition (LI) and spike frequency adaptation (SFA)—on odor learning under varying noise conditions in the fly olfactory circuit.

Our study is positioned at the interface of computational neuroscience and spiking neural networks, utilizing AI methods to unravel learning processes in the fly olfactory circuit. This interdisciplinary nature, while enriching, also inherits some complex challenges that extend beyond the scope of a single work or the timeframe of the rebuttal period from both fields, such as biologically plausible learning mechanisms. It is widely accepted that BPTT is more efficient compared to STDP; however, to our knowledge, the latter mechanism has not yet been proven to occur in vivo during learning.

During the rebuttal phase, we have conducted extensive new experiments and analyses to address the reviewers' questions, including:
*   Demonstrating the synergistic effects of lateral inhibition and spike frequency adaptation, which significantly enhances odor discrimination.
*   Comprehensively comparing fly olfactory circuit model with unstructured networks, highlighting the advantages of its inherent fixed sparse connectivity.
*   Verifying the robustness of our findings across different noise types (Gaussian and Ornstein-Uhlenbeck) and through comprehensive parameter sensitivity analyses.
*   Providing detailed justifications for our modeling choices, such as the use of SNNs and time-averaged MBON outputs, and clarifying model notations and figure explanations to improve clarity.

We acknowledge that certain complex aspects, inherent to this interdisciplinary field and requiring extensive investigation, could not be fully resolved in this single work, particularly regarding the learning algorithms: While BPTT was used to establish an upper bound on architectural performance, the exact learning mechanisms in biological circuits remain unknown and controversial. Future work will explore more biologically plausible alternatives.

All the new analysis and limitations will be included in our final manuscript.

---

### Decision · Program_Chairs · 2025-09-17

**Decision:**

Accept (poster)

**Comment:**

This paper is a borderline submission that was aided by a robust discussion period that led to new results and cleaner explanations, which mostly satisfied the reviewers. While there was still discussion about whether the results would prove impactful enough, I believe this is a solid addition to NeurIPS, where the rigorous computational work here could inspire future Neuroscience experiments.